# Efficacy and Safety of Neoadjuvant Chemotherapy Combined with Adjuvant Chemotherapy for Locally Advanced Colon Cancer: A Propensity Score-Matching Analysis

**DOI:** 10.3390/medicina58111505

**Published:** 2022-10-22

**Authors:** Wei Zeng, Yi Liu, Chuandong Wang, Changshun Yang, Shengtao Lin, Weihua Li

**Affiliations:** 1Shengli Clinical Medical College of Fujian Medical University, Fuzhou 350001, China; 2Department of Endoscopy, National Cancer Center/National Clinical Research Center for Cancer/Cancer Hospital, Chinese Academy of Medical Science and Peking Union Medical College, Beijing 100021 China; 3Department of Surgical Oncology, Fujian Provincial Hospital, Fuzhou 350001, China

**Keywords:** neoadjuvant chemotherapy (NAC), locally advanced colon cancer (LACC), preoperative treatment, propensity score matching, survival

## Abstract

*Background and Objectives*: Increasing evidence supports the use of neoadjuvant chemotherapy (NAC) for locally advanced colon cancer (LACC). However, its effectiveness remains controversial. This study explored the safety and efficacy of NAC combined with laparoscopic radical colorectal cancer surgery and adjuvant chemotherapy (AC) for LACC. *Materials and Methods*: We retrospectively analyzed 444 patients diagnosed with LACC (cT4 or cT3, with ≥5 mm invasion beyond the muscularis propria) in our hospital between 2012 and 2015. Propensity score matching (PSM; 1:2) was performed to compare patients treated with NAC and those treated with adjuvant chemotherapy (AC). *Results*: Overall, 42 patients treated with NAC were compared with 402 patients who received only AC. After PSM, 42 patients in the NAC group were compared with 84 patients in the control group, with no significant differences in the baseline characteristics between groups. The pathological tumor sizes in the NAC group were significantly smaller than those in the AC group (3.1 ± 2.1 cm vs. 5.8 ± 2.5 cm). Patients in the NAC group had a significantly lower T stage than those in the AC group (*p* < 0.001). After neoadjuvant chemotherapy, a significant response was observed in four (9.6%) patients, with two (4.8%) showing a complete response. The 5-year overall survival rates (88.1% vs. 77.8%, *p* = 0.206) and 5-year disease-free survival rates (75.1% vs. 64.2%, *p* = 0.111) did not differ between the groups. However, the 5-year cumulative rate of distant recurrence was significantly lower in the NAC than in the AC group (9.6% vs. 29.9%, *p* = 0.022). *Conclusions*: NAC, combined with AC, could downstage primary tumors of LACC and seems safe and acceptable for patients with LACC, with a similar long-term survival between the two treatments.

## 1. Introduction

Colon cancer (CC) is the fourth most common type of cancer worldwide [1]. Among patients with CC, a substantial proportion that presents with locally advanced colon cancer (LACC) (T4 or T3, with ≥5 mm invasion beyond the muscularis propria) still have an unsatisfactory prognosis, with 5-year survival rates ranging from 55% to 88%, despite developments in surgical technique and chemotherapy regimens [2]. Worldwide, the current standard treatment strategy for LACC is radical surgical resection of the tumor (R0 resection), followed by adjuvant chemotherapy. Regarding the clinical treatment strategy of other solid tumors, neoadjuvant chemotherapy (NAC) has been successfully applied in the clinical treatment of cancers, including rectal and breast cancer [3,4,5]. Relevant research has suggested that neoadjuvant chemotherapy (NAC) was useful in promoting a reduction in tumor burden prior to surgery and the eradication of micro-metastases [6], which achieved a higher rate of R0 resection. However, it usually takes about one month for patients to fully recover from surgery and receive adjuvant chemotherapy (AC). A previous study recognized that metabolic activity increased after surgical removal of the primary tumor, suggesting that surgical stimulation of growth factors may be one of the factors promoting postoperative metastasis [7]. From this perspective, preoperative NAC may have a positive impact on patient prognosis [8]. Therefore, new treatment strategies urgently need to be proposed and validated.

However, there are still relatively few studies showing the usefulness of NAC for survival in patients with LACC. Most recent studies have focused on demonstrating the feasibility and safety of NAC [9,10,11]. The FOxTROT study showed that preoperative chemotherapy combining 5-fluorouracil and oxaliplatin with or without panitumumab in patients with resectable T4 or T3 colon cancer had a significant effect on tumor down-staging and had high safety [12]. The ongoing French clinical trial PRODIGE 22-ECKINOXE and the Chinese COLARC study have both confirmed that NAC is feasible, with acceptable tolerability, but is not associated with an increased major pathological response rate [13,14,15]. A retrospective study also showed that patients with clinical T4b CC treated with NAC might have an improved survival rate [16], but this has not been observed in patients with clinical T3 or T4a CC. In terms of clinical guidelines, the current National Comprehensive Cancer Network (NCCN) guidelines also recommend preoperative neoadjuvant chemoradiotherapy as an option for patients with initially unresectable, non-metastatic T4 colon cancer [17].

The application of NAC for LACC is challenged by concerns that patients may lose the opportunity to undergo radical surgery due to the progression of the primary tumor during NAC, while some patients may receive over-treatment due to inaccurate computed tomography (CT) staging. With the advancements in CT, many studies have confirmed the accuracy of CT technology in staging CC [18,19,20], and CT scanning can accurately identify high-risk (T3/4) colon cancers with minimal over-staging of T1/T2 tumors [21]. Given the potential advantages and disadvantages, we conducted this study to investigate the perioperative efficacy and postoperative outcomes to evaluate whether NAC could improve prognosis in patients with LACC and who only received surgery combined with postoperative AC for the time being.

## 2. Materials and Methods

### 2.1. Patient Selection

We reviewed the data of 444 patients with LACC (T4 or T3, with ≥5 mm invasion beyond the muscularis propria) who underwent surgery at the Fujian Provincial Hospital between 2012 and 2015 (Figure 1). Patients were randomly assigned to two groups: 42 received preoperative NAC combined with AC, while the remaining 402 received postoperative AC. The inclusion criteria were as follows: (1) histologically confirmed colon cancer; (2) CT-verified colon cancer at clinical stage T4a or T3, with ≥5 mm invasion beyond the muscularis propria; and (3) radical surgery. The exclusion criteria were as follows: (1) distant metastasis detected upon preoperative examination; (2) simultaneous malignancies from other organs or prior malignancy; (3) serious cardiovascular or cerebrovascular diseases, liver and kidney dysfunction, severe blood system diseases, immune system diseases, or severe mental disorder; (4) incomplete or inaccurate medical records; and (5) below 18 years old or over 90 years old at the time of diagnosis. This study was reviewed and approved by the Ethics Committee of the Fujian Provincial Hospital and was registered under the ethics committee approval number K2017-09-070. All data were anonymized, and the requirement for informed consent was therefore waived. All study procedures were performed in accordance with the Helsinki Declaration of 1964 and its later versions.

### 2.2. Treatment Regimes

Initial clinical staging using colonoscopy with biopsy confirmation and abdominal computed tomography (CT) was performed in all cases. Patients in the NAC group received 6 cycles of XELOX (capecitabine 1000 mg/m^2^ orally days 1–14 q3w, oxaliplatin 130 mg/m^2^ iv day 1 q3w) after diagnosis and underwent radical surgery three weeks after the last cycle of NAC. The response to NAC was assessed every three cycles by performing a CT scan (according to RECIST [22]) and measuring serum carcinoembryonic antigen (CEA) and carbohydrate antigen199 (CA199) levels. Further AC was determined based on the pathological results and the patient’s willingness to undergo the remaining two cycles of XELOX. For patients in the AC group, radical surgery was performed first after diagnosis, and patients received eight cycles of AC (XELOX, capecitabine 1000 mg/m^2^ orally days 1–14 q3w, oxaliplatin 130 mg/m^2^ iv day 1 q3w), depending on the histological stage and the pathological response. Follow-up was performed 1 month after surgery, every 3 months for 3 years, every 6 months for 5 years, and yearly thereafter.

### 2.3. Data Collection

The following variables were included in the analysis: sex, age, tumor site, tumor size, gross type, tumor differentiation, histopathology, clinical T and N stages, serum CEA and CEA levels, ASA grade, body mass index (BMI), operation time, estimated blood loss, time to start the diet, length of hospital stay, toxic effect, pathologic outcomes according to the American Joint Committee on Cancer (AJCC) guidelines (8th edition), and postoperative morbidity and mortality. Toxicity was assessed according to the Common Toxicity Criteria for Adverse Events (version 3.0). The date of diagnosis was defined as the date of the first histological confirmation of malignancy, most often the day of the endoscopic biopsy. After resection, the pathologist performed the final stage. A pathological tumor (ypT) and nodal staging were compared with clinical staging in both groups to assess the downstaging effects of neoadjuvant CT. R0 resection was achieved if the resection margins were microscopically tumor-free. In the case of irradical resection, the resection was either labeled R1 (microscopic involvement of the resection margins) or R2 (macroscopic involvement). Major postoperative complications such as wound infection, ileus, and anastomotic leakage were recorded. The primary outcome was overall survival. The secondary endpoints were recurrence rate, disease-free survival (DFS), and chemotherapy toxicity.

### 2.4. Statistical Analysis

The chi-square test or Fisher’s exact probability method was used to compare classified variables between the two groups. An independent-samples *t*-test was used to compare normally distributed continuous variables. Nonparametric Mann–Whitney U tests were applied when the variance was not normally distributed. Propensity score matching was applied to reduce the possibility of selection bias and adjust for significant differences in the baseline characteristics of the patients. The propensity score was calculated based on sex, age, tumor site, tumor size, gross type, tumor differentiation, histopathology, clinical T and N stage, CEA levels, American Society of Anesthesiologists (ASA) grade, and body mass index (BMI). Patients in the NAC group were matched 1:2 using nearest neighbor matching based on the closest propensity score to those in the AC group. Overall survival and disease-free survival rates were estimated using the Kaplan–Meier method. Statistical significance was set at *p* < 0.05. Statistical analysis was performed using SPSS for Windows (version 22.0; SPSS Inc., Chicago, IL, USA).

## 3. Results

### 3.1. Baseline Characteristics

This study included 444 patients with LACC who underwent radical surgical resection between January 2012 and June 2015. Among them, 42 patients received NAC before surgery, while the remaining 402 patients underwent surgical resection without preoperative chemotherapy. Before propensity score matching, sex, gross type, tumor differentiation, histopathology, cT and cN stages, serum CEA level, serum CA199 level, ASA, and BMI were not significantly different between the groups (Table 1). However, compared to the AC group, patients in the NAC group were significantly older (66.48 ± 11.98 years vs. 61.60 ± 13.68 years, *p* = 0.027), the tumor sizes were significantly larger (5.0 ± 1.7 vs. 4.2 ± 2.0 cm, *p* = 0.009), and more tumors were located in the left colon (*p* = 0.015). A propensity score was calculated to adjust for biases caused by differences in baseline characteristics between the two groups. After matching, there were no significant differences in any baseline characteristics between the groups (Table 1). We found that several indices of patients in the NAC group, such as body mass index (BMI), serum CEA level, serum CA199 level, and American Society of Anesthesiologists (ASA) score, improved after six cycles of NAC. (Table 2, *p <* 0.05).

### 3.2. Perioperative Outcomes

The operation time, estimated blood loss, time to bowel movement, time to a liquid diet, time to a soft diet, postoperative hospital stays, and complications within 30 days of surgery were similar between the two groups (Table 3). In addition, there was no significant difference in mortality between the two groups 30 days after surgery. Regarding the toxic effects of chemotherapy, there was no significant difference in the incidence of gastrointestinal, hematologic, and dermatologic effects; however, the NAC group had a lower incidence of any grade 3 or 4 toxic effects than the AC group (10.0% vs. 25.9%, *p* = 0.041). Four (9.5%) and eighteen (21.4%) patients did not complete the full cycles of chemotherapy in the NAC and AC groups due to toxic effects, respectively.

### 3.3. Pathological Outcomes

None of the patients experienced progression during neoadjuvant chemotherapy, and the NAC group achieved a smaller tumor size than the AC group (3.1 ± 2.1 vs. 5.8 ± 2.5 cm, *p* < 0.001). In all patients, the cT stage was reported before the start of NAC. Four patients showed significant downstaging of the primary tumor after systemic therapy (cT3-4 to pT0-2, 9.5%), while two patients showed a complete pathological response (pT0; Table 4). None of the patients in the NAC group had nodal over-staging. Although only three patients (21.4%) were diagnosed with cN1and finally had pN2 disease, up to 15 patients (65.2%) were diagnosed with cN0 and finally had pN1-2 disease (Table 5).

### 3.4. Survival

The median follow-up periods in the NAC and AC groups were 56 (12–80) and 66.5 (2–83) months, respectively, while the corresponding 5-year overall survival rates were 88.1% and 77.8%, respectively. This difference was not significant (*p* = 0.206; Figure 2a). Furthermore, there was no significant difference in the 5-year progression-free survival between the two groups (75.1% vs. 64.2%, *p* = 0.111; Figure 2b), nor in the incidence of 5-year local recurrence (18.3% vs. 15.3%; *p* = 0.935; Figure 2c). However, the 5-year cumulative incidence of distal recurrence was 9.6% in the NAC group, which was significantly lower than that in the AC group (29.9%, *p* = 0.018; Figure 2d).

## 4. Discussion

NAC for LACC has become increasingly frequently applied in the clinic; however, its applicability remains controversial [6,9,11,16]. Our study illustrated that NAC combined with AC was not only safe but also resulted in significant tumor downstaging in patients with LACC, and the long-term outcomes were similar to those of patients who underwent surgery directly after diagnosis.

It is well known that chemotherapy drugs induce certain toxicity towards the liver and kidney, and it has been suggested that NAC may be associated with unnecessary patient morbidity due to chemotherapeutic toxicities. When comparing patients who received NAC with those who did not, we found that patients in the NAC group had a lower incidence of grade 3 or 4 toxicities. Clinically, neoadjuvant therapy toxicity (grade 3 or 4) was observed in only 10% of the patients in the NAC group. Interestingly, our further research suggested that several indices of patients in the NAC group, such as serum CEA level, serum CA199 level, BMI, and ASA score, were improved after six cycles of NAC. This may be because NAC was usually carried out before radical surgery, delaying the operation time by about 12–18 weeks. In addition, we increased the nutritional intake through enteral and parenteral nutrition during chemotherapy. As a result, patients had the opportunity to improve their physical condition before surgery. Patients tolerated b preoperative chemotherapy better than postoperative adjuvant chemotherapy because they were in a relatively healthier state.

A previous study indicated that the rates of adverse reactions and surgical complications did not differ between patients who underwent NAC and those who did not. Karoui et al. and the FOxTROT Collaborative Group both demonstrated that there was no significant difference in postoperative anastomotic leaks, wound infections, or return to the theater between the neoadjuvant and control arms in both RCTs [12,15]. The results of our study suggest that operation time, estimated blood loss, time to bowel movement, time to a liquid diet, time to a soft diet, postoperative hospital stays, and mortality within 30 days of surgery did not show any statistical difference between the NAC and AC groups. In addition, the occurrence of major complications, such as wound infection, ileus, and anastomotic leakage, was equal between the groups. The results of this study clearly showed that NAC is well tolerated with an acceptable side effect profile for an average of less than 30 days after surgery. Thus, we supposed that NAC was non-inferior in terms of safety and did not increase surgical complications or mortality compared to standard surgery. Other concerns raised about NAC were related to the possibility that the response of tumors to neoadjuvant therapies remains variable; a subgroup of patients may not achieve any downstaging of the tumor, and some of them may even show disease progression, as observed in locally advanced rectal cancer due to delays in operative intervention [23,24]. However, it was encouraging that no progression was observed during neoadjuvant chemotherapy in this study. This finding is in agreement with the results of a previous study [12].

The subjects included in our present study were patients with LACC, such as T4 or high-risk T3 (with ≥5 mm invasion beyond the muscularis propria), without distant metastases. Identifying this patient population relied heavily on accurate CT staging, as it guided the need for neoadjuvant therapy. CT staging was found to be accurate, with an overall sensitivity of 90% in detecting tumor invasion beyond the bowel wall and nodal involvement in a previous meta-analysis [25]. In this study, we enrolled 23 patients diagnosed with the cT4 stage in the NAC group. Tumor grade regression of the specimen is an important factor directly related to chemotherapy response [26,27]. The results of our study showed that the benefits of NAC included the significant downsizing of the primary tumor and downstaging of the T stage. However, several studies have indicated that the complete pathological response rate of LACC was between 2–4.6%, which is significantly lower than that of rectal cancers, which ranged from 15% to 25% [10,28,29]. In our study, the sizes of primary tumors were markedly reduced after NAC in 42 patients. Additionally, evidence of significant downstaging (cT3-4 to pT0-2, 9.5%) was demonstrated in 9.5% of patients, and a complete pathological response (pT0) was observed in 4.8% of patients. This is also in agreement with the results of a previous study [11].

Recently, a systematic review and meta-analysis concluded that NAC could significantly improve disease-free survival and overall survival in patients with rectal cancer [30]. Similarly, Cheong et al. found that patients with colon cancer receiving NAC also had better overall survival and disease-free survival [31]. In contrast, several studies have suggested that the overall survival of patients receiving NAC was similar to that of patients without NAC [11,16]. Our research showed no significant difference in overall survival and disease-free survival between the NAC and AC groups; however, NAC significantly reduced the incidence of distant recurrence. This may be explained by the fact that circulating tumor cells and lymph node metastasis could be eradicated by early systemic NAC. Furthermore, NAC may shrink tumors and reduce tumor cell shedding caused by surgical trauma. A related study showed that surgery stimulates growth factors and induces immunosuppression, which may promote tumor progression and the spread of micrometastases in the postoperative setting [13]. Surgery after NAC can remove the tumor more radically and eradicate systemic micrometastases earlier. This may prevent the occurrence of distant relapses.

Despite these positive findings, this study had several limitations. First, the sample size was relatively small. Second, selection bias could have occurred in the control group because only patients who were able to undergo adjuvant CT were included, and patients who died postoperatively or had severe complications were excluded. Third, although propensity score matching was performed to balance the significant baseline characteristics of patients, RCTs nevertheless need to be conducted to confirm our results.

## 5. Conclusions

Overall, our findings showed that a lower incidence of any grade 3 or 4 toxic effects were observed in the NAC group, and there was no significant increase in postoperative complications or mortality. NAC combined with AC could be used to downstage the primary tumor of the LACC and eliminate potential micrometastases. NAC combined with AC appears to be a safe and acceptable modality for patients with LACC. However, additional large randomized trials with longer follow-up times are needed to provide more reliable results.

## Figures and Tables

**Figure 1 medicina-58-01505-f001:**
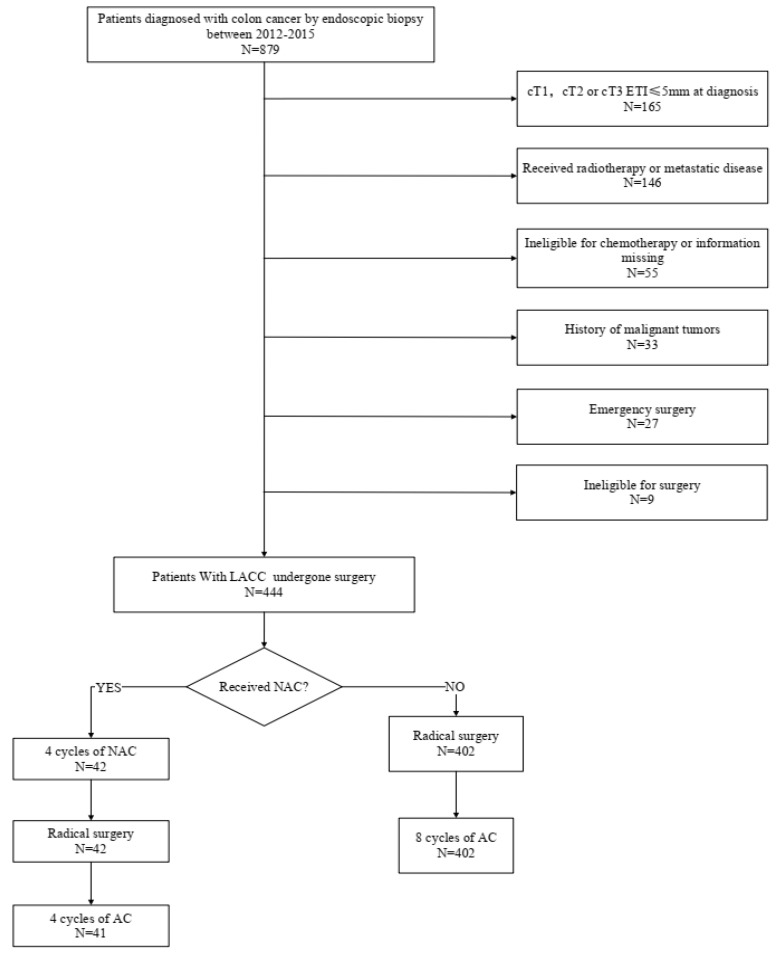
Flowchart of patient selection.

**Figure 2 medicina-58-01505-f002:**
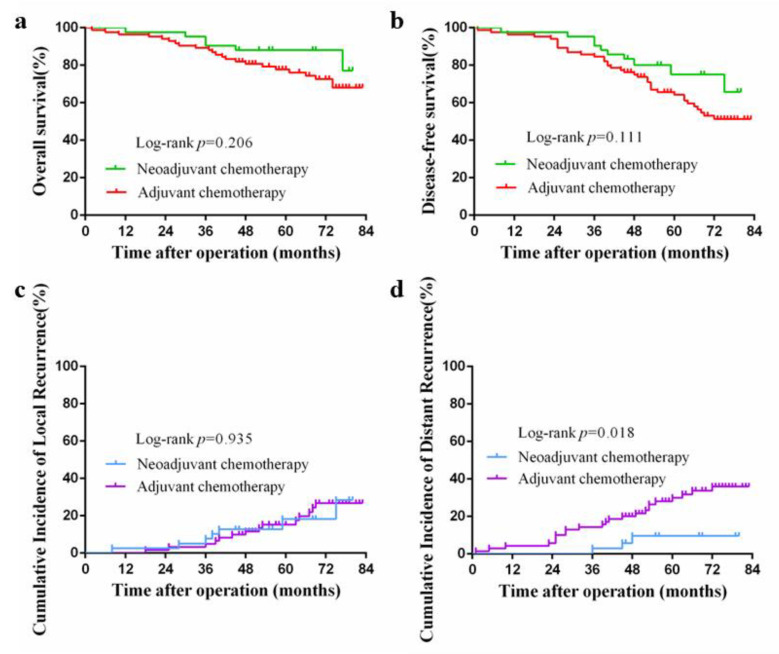
(**a**) Kaplan–Meier curve for overall survival after propensity score matching; (**b**) Kaplan–Meier curve for disease-free survival after propensity score matching; (**c**) Kaplan–Meier curve for cumulative incidence of local recurrences; (**d**) Kaplan–Meier curve for cumulative incidence of distant recurrences.

**Table 1 medicina-58-01505-t001:** Overall patient and tumor characteristics before and after PSM for the NAC and AC groups.

Variable	Raw Data		*p*	After Propensity Matching		*p*
	NAC (*n* = 42)	AC (*n* = 402)		NAC (*n* = 42)	AC (*n* = 84)	
Gender			0.612			0.377
Male	22	227		22	37	
Female	20	175		20	47	
Age, years			0.027			0.670
Mean ± SD (range)	66.48 ± 11.98	61.60 ± 13.68		66.48 ± 11.98	65.51 ± 11.95	
Tumor site			0.015			0.777
Right colon	12	194		12	22	
Left colon	30	208		30	62	
Tumor size, cm (imaging)			0.009			0.635
Mean ± SD	5.0 ± 1.7	4.2 ± 2.0		5.0 ± 1.7	4.8 ± 2.0	
Morphology			0.855			0.077
Infiltrative	1	23		1	0	
Ulcerative	27	247		27	43	
Expanding	14	132		14	41	
Tumor differentiation			0.887			0.280
Well or moderately	34	329		34	74	
Poorly, others	8	73		8	10	
Histopathology			0.847			0.541
Tubular adenocarcinoma	34	328		34	74	
Mucinous adenocarcinoma	6	55		6	7	
Signet ring cell carcinoma	0	6		0	1	
Others	2	13		2	2	
cT stage *			0.436			0.172
cT3	19	157		19	27	
cT4	23	245		23	57	
cN stage *			0.214			0.591
cN0	20	213		20	38	
cN1	10	111		10	28	
cN2	9	69		9	15	
cNx	3	9		3	3	
CEA, ng/ml			0.093			0.074
Median (P_25_, P_75_)	5.62 (2.81, 13.63)	4.94 (2.08, 13.10)		5.62 (2.81, 13.63)	4.82 (2.00, 12.68)	
CA199, U/mL			0.328			0.362
Median (P_25_, P_75_)	19.63 (11.97, 27.75)	17.02 (8.38, 27.84)		19.63 (11.97, 27.75)	16.10 (10.03, 27.40)	
ASA			0.333			0.960
I	26	203		26	53	
II	13	126		13	24	
III	3	67		3	7	
IV	0	6		0	0	
BMI			0.269			0.153
Mean ± SD (range)	20.64 ± 4.37	21.51 ± 4.88		20.64 ± 4.37	21.85 ± 4.49	

* According to the AJCC Cancer Staging Manual, 8th edition. Abbreviations: ASA, American Society of Anesthesiologists; BMI, body mass index.

**Table 2 medicina-58-01505-t002:** Patient characteristics after neoadjuvant chemotherapy for the NAC group.

Variable	NAC-Before (*n* = 42)	NAC-After (*n* = 42)	*p*
CEA, ng/mL			0.003
Median (P_25_, P_75_)	5.62 (2.81, 13.63)	3.25 (2.66, 4.20)	
CA199, U/ml			0.001
Median (P_25_, P_75_)	19.63 (11.97, 27.75)	12.42 (4.55, 1.40)	
ASA			0.009
I	26	37	
II	13	5	
III	3	0	
IV	0	0	
BMI			0.001
Mean ± SD	20.65 ± 4.37	23.46 ± 3.28	

Abbreviations: NAC-after, patients of the NAC group after 6 cycles of neoadjuvant chemotherapy; NAC-before, patients of the NAC group before neoadjuvant chemotherapy; ASA, American Society of Anesthesiologists; BMI, body mass index.

**Table 3 medicina-58-01505-t003:** Comparison of perioperative outcomes between NAC and AC groups.

Variable	NAC (*n* = 42)	AC (*n* = 84)	*p*
Operation time, min			0.183
Median (P_25_, P_75_)	185.50 (165.50, 201.00)	175.50 (147.25, 200.75)	
Estimated blood loss, ml			0.111
Median (P_25_, P_75_)	50.00 (35.00, 60.00)	55.00 (40.00, 65.00)	
Anal exhaust time, day			0.757
Median (P_25_, P_75_)	3.00 (2.00, 3.00)	3.00 (2.00, 3.00)	
Time to liquid diet, day			0.375
Median (P_25_, P_75_)	2.00 (1.00, 2.00)	1.50 (1.00, 2.00)	
Time to soft diet, day			0.383
Median (P_25_, P_75_)	3.00 (3.00, 4.00)	4.00 (3.00, 4.00)	
Postoperative hospital stays, day			0.419
Median (P_25_, P_75_)	6.00 (5.00, 8.00)	6.00 (5.00, 7.00)	
Complication within 30 days of surgery			1.000
None	33	67	
Wound infection	2	5	
Ileus	5	9	
Anastomotic leakage	2	3	
Mortality within 30 days of surgery			0.552
No	42	82	
Yes	0	2	
Toxic effect *			
Gastrointestinal **	7	19	0.436
Hematologic effects	10	32	0.109
Dermatologic effects	9	25	0.321
Any grade 3 or 4 toxic effect	4	22	0.041

* According to National Cancer Institute Common Toxicity Criteria; ** Nausea, vomiting, and diarrhea.

**Table 4 medicina-58-01505-t004:** Comparison of pathologic outcomes between the NAC and AC groups.

Variable	NAC (*n* = 42)	AC (*n* = 84)	*p*
Tumor size, cm (pathological)			<0.001
Mean ± SD	3.1 ± 2.1	5.8 ± 2.5	
T stage *			<0.001
T0	2	0	
T1	2	0	
T2	8	0	
T3	23	29	
T4	7	55	
N stage *			0.310
N0	22	49	
N1	10	24	
N2	10	11	
Resection margin			1.000
R0	40	81	
R1	2	3	
Angiolymphatic invasion			0.725
Positive	35	72	
Negative	7	12	
Nerve invasion			1.000
Positive	40	81	
Negative	2	3	

* According to the AJCC Cancer Staging Manual, 8th edition.

**Table 5 medicina-58-01505-t005:** Clinical and pathological nodal staging.

**(a) Nodal downstaging in patients who received NAC**				
**(a)**				
	**Pathological N-score**			
	**pN0**	**pN1**	**pN2**	**Total**
Clinical N-score				42
cN0	20	0	0	20
cN1	0	10	0	10
cN2	0	0	9	9
cNx	2	0	1	3
Total	22	10	10	42
**(b) Comparison of clinical and pathological nodal staging in patients treated with AC.**				
**(b)**				
	**Pathological N-score**			
	**pN0**	**pN1**	**pN2**	**Total**
Clinical N-score				
cN0	23	9	6	38
cN1	14	11	3	28
cN2	10	3	2	15
cNx	2	1	0	3
Total	49	24	11	84

Abbreviations: NAC, neoadjuvant chemotherapy; AC, adjuvant chemotherapy.

## Data Availability

Please contact the corresponding author (Weihua Li, email: liwh@fjmu.edu.cn) for data requests.

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
