# Peer review of "Efficacy and Safety of Neoadjuvant Chemotherapy Combined with Adjuvant Chemotherapy for Locally Advanced Colon Cancer: A Propensity Score-Matching Analysis"

_medicina, 2022, doi:10.3390/medicina58111505_

Round 1

Reviewer 1 Report

Ethics committee approval registration number is required – further development is required as to the acceptance of patients to receive NAC versus AC.

Criteria of differentiations is required between the two groups – why 402 received AC and 42 received NAC. The authors suggest tumor size as on of the selection criteria between the two groups, but it requires further explanation. Propensity-matching do not substitute clear inclusion and exclusion criteria in a retrospective study.

 The premises of the study is ill explained and unclear leading to a misunderstanding of the point of the article. There is a huge gap in sample size that is not explained by inclusion / exclusion criteria or treatment protocol.

 The entire article has numerous grammatical errors, punctuation errors, formatting errors, different fonts used in different sections.

Author Response

Response to Reviewer 1 Comments

  1. Ethics committee approval registration number is required – further development is required as to the acceptance of patients to receive NAC versus AC.

Response: Thank you for your comments on this study. The ethics committee approval registration number is K2017-09-070, we have added more details in this section and the relevant files were uploaded. (Line 91-92)

  1. Criteria of differentiations is required between the two groups – why 402 received AC and 42 received NAC. The authors suggest tumor size as on of the selection criteria between the two groups, but it requires further explanation. Propensity-matching do not substitute clear inclusion and exclusion criteria in a retrospective study.

Response: Patients were randomly assigned to groups, as the reviewer suggested, we have added more details in this section. (Line 81) The difference between the two groups may be due to the insufficient number of patients enrolled in neoadjuvant therapy at that time. Clinicians tended to apply neoadjuvant therapy in patients with larger tumor size. So, we use the PSM to balance the groups differences.

  1.  The premises of the study is ill explained and unclear leading to a misunderstanding of the point of the article. There is a huge gap in sample size that is not explained by inclusion / exclusion criteria or treatment protocol.

Response: Patients were randomly assigned to groups, as the reviewer suggested, we have added more details in this section. (Line 81) The difference between the two groups may be due to the insufficient number of patients enrolled in neoadjuvant therapy at that time. Clinicians tended to apply neoadjuvant therapy in patients with larger tumor size. So, we use the PSM to balance the groups differences.

  1. The entire article has numerous grammatical errors, punctuation errors, formatting errors, different fonts used in different sections.

Response: Thank you for your comments on this study. When we received your suggestion, we immediately asked our colleague who is native English speaker to help us revise it. We really hope that this case can be accepted.

Reviewer 2 Report

General comments:

This study showed the efficacy of neoadjuvant chemotherapy (NAC) over adjuvant chemotherapy (AC) in tumor downsizing, downstaging, and lower incidence of distant metastasis in patients with cT3 or T4a colorectal cancer.

The followings are concerns to be reflected in the revised manuscript.

Specific comments:

1.      The authors defined “NAC” as preoperative XELOX treatment for 6 cycles. Moreover, the authors mentioned that further AC was determined on pathological results and patient's willingness to receive the remaining 2 cycles of XELOX. However, Figure 1 shows that NAC patients had received 4 cycles of XELOX preoperatively and 41 out of 42 patients had received 4 cycles of XELOX. This treatment may not be “NAC” alone, rather “combined NAC and AC”. Given that postulation, the title, methods, results, conclusions may be modified accordingly.

2.      The authors found that NAC showed advantages in tumor downsizing, downstaging, and reduced distant metastasis whereas it did not show benefits in local control or overall survival. These results may limit the advantages of NAC (with AC) compared with AS alone.

3.      “NAC could eliminate potential micro-metastases” is not supported by the results obtained in this study. Therefore, this proposition may better be removed from the abstract and can be remained in the discussion with some supportive evidences.

4.      Please describe the indications of NAC, AC, or neither of both in the methods.

5.      It seems very unnatural that ASA-PS and BMI had significantly improved after receiving NAC. These results cannot be explained by “sufficient” interval between the NAC and surgery.

6.      Please specify the source of “grade” of adverse events in the main text (not only in the table).

7.      Information about the completion rate of NAC and AC are valuable.

Author Response

Response to Reviewer 2 Comments

  1. The authors defined “NAC” as preoperative XELOX treatment for 6 cycles. Moreover, the authors mentioned that further AC was determined on pathological results and patient's willingness to receive the remaining 2 cycles of XELOX. However, Figure 1 shows that NAC patients had received 4 cycles of XELOX preoperatively and 41 out of 42 patients had received 4 cycles of XELOX. This treatment may not be “NAC” alone, rather “combined NAC and AC”. Given that postulation, the title, methods, results, conclusions may be modified accordingly.

Response: Thank you for your comments on this study. As the reviewer suggested, we have modified relevant parts in our article.

  1. The authors found that NAC showed advantages in tumor downsizing, downstaging, and reduced distant metastasis whereas it did not show benefits in local control or overall survival. These results may limit the advantages of NAC (with AC) compared with AS alone.

Response: Studies on NAC have shown mixed results in terms of long-term survival. The relevant content has been explained in this article. (Line 318-322) However, the benefits of neoadjuvant therapy cannot be ignored, such as lower incidence of toxic effect and reduced distant recurrences rates.

  1. “NAC could eliminate potential micro-metastases” is not supported by the results obtained in this study. Therefore, this proposition may better be removed from the abstract and can be remained in the discussion with some supportive evidences.

Response: As the reviewer suggested, we have corrected this part in our article.

  1. Please describe the indications of NAC, AC, or neither of both in the methods.

Response: Patients were randomly assigned to groups, as the reviewer suggested, we have added more details in this section. (Line 81)

  1. It seems very unnatural that ASA-PS and BMI had significantly improved after receiving NAC. These results cannot be explained by “sufficient” interval between the NAC and surgery.

    Response: We increased the nutritional intake through enteral and parenteral nutrition during chemotherapy. As a result, patients could have the opportunity to improve their physical conditions before surgery. We have added more details in this section. (Line 278-279)

  1. Please specify the source of “grade” of adverse events in the main text (not only in the table).

Response: As the reviewer suggested, we have added more details in this section. (Line 123-124)

  1. Information about the completion rate of NAC and AC are valuable.

Response: There were 4 (9.5%) and 18 (21.4%) didn’t complete the full cycles of chemotherapy in the NAC group and AC group due to toxic effects respectively. As the reviewer suggested, we have added more details in this section. (Line 220-221)

Reviewer 3 Report

I think this study is well written and interesting. What I cannot see clearly explained are the reasons/indications for the neoadjuvant chemotherapy compared with the adjuvant chemotherapy group.

The entire study and results and conclusions are based on this separation of groups, yet without why the choice of NAC or AC, a reader cannot interpret anything about this data or repeat this study, or benefit from its findings.

Author Response

Response to Reviewer 3 Comments

  1. I think this study is well written and interesting. What I cannot see clearly explained are the reasons/indications for the neoadjuvant chemotherapy compared with the adjuvant chemotherapy group.The entire study and results and conclusions are based on this separation of groups, yet without why the choice of NAC or AC, a reader cannot interpret anything about this data or repeat this study, or benefit from its findings.

Response: Thank you for your comments on this study. Patients were randomly assigned to groups, as the reviewer suggested, we have added more details in this section. (Line 81) The difference between the two groups may be due to the insufficient number of patients enrolled in neoadjuvant therapy at that time. Clinicians tended to apply neoadjuvant therapy in patients with larger tumor size. So, we use the PSM to balance the groups differences.

Round 2

Reviewer 1 Report

Good for publishing. 

Reviewer 2 Report

The revised manuscript has been well corrected in accordance with the reviewers' suggestions.

Reviewer 3 Report

All of my concerns have been answered. Paper is easier to now understand.